# Importance of Perception of Errors and Challenges for Improving Psychological Conditioning: Mediating Effect of Expectancy-Value Using the Phantom Model for Taekwondo Athletes

**DOI:** 10.3390/ijerph19106112

**Published:** 2022-05-17

**Authors:** Young Kyun Sim, Hak Hwan Kim, Joon Ha Shin, Eun Chul Seo, Min-Seong Ha

**Affiliations:** 1Department of International Sports, Dankook University-Chungcheongnam-do, 119 Dandae-ro, Dongnam-gu, Cheonan-si 31116, Korea; simco76@dankook.ac.kr (Y.K.S.); a982jh@naver.com (J.H.S.); 2Department of Sports Health Regeneration, Cheong Ju University-Chungcheongbuk-do, 298 Daeseong-ro, Cheongwon-gu, Cheongju-si 28503, Korea; hakbongs@hanmail.net; 3Department of Physical Education, Wonkwang University-Iksan, 460 Iksan-daro, Iksan 54538, Korea; 4Department of Sports Culture, College of the Arts, Dongguk University-Seoul, 30 Pildong-ro 1-gil, Jung-gu, Seoul 04620, Korea

**Keywords:** psychological conditioning, perception of error, expectancy value, challenge, structural equation modeling, athlete, sports

## Abstract

Maintenance of positive psychological conditioning can be a key factor in eliciting high human performance. In particular, perception of error forms a causal relationship with challenges regarding task performance. Therefore, this study aimed to investigate the mediation effect of expectancy value in the relationship between the perception of error and challenge using the Phantom Model through quantitative research. This study analyzed the causal relationship between perception of error, expectancy value, and challenge in 423 young Taekwondo athletes. Frequency analysis, confirmatory factor analysis, correlation analysis, and structural equation modeling were performed on the collected data using Jamovi 1.0.1 and AMOS 23.0 statistical programs to verify the hypothesis. The challenge demonstrated a significant difference in relation to the perception of error. These results demonstrated that the perception of error not only directly affected one’s challenge but also explained the increased challenge by mediating expectations for success and subjective values. Hence, the positive perception of error increased the athletes’ expectancy value and challenge.

## 1. Introduction

Mistakes are perceived negatively because they cause negative emotions that require continuous adjustment [1]. Recently, mistakes were reported as a concomitant process for future growth, and the importance of attitudes toward mistakes and the method of interpreting mistakes were highlighted [2,3,4]. Perception of error, which is presented as a multidimensional concept (cognitive, emotional, behavioral), refers to an individual’s cognitive and motivational tendency to perceive and accept mistakes [3]. According to the error-driven learning theory, mistakes are inevitable in the learning process [5,6], although using them positively can improve learning outcomes. Moreover, evidence supporting the justification for error perception in sports is presented [4,7].

Mistakes that occur in sports have been suggested to directly or indirectly affect the players’ cognitive and physical anxiety, negative emotions, and optimal level of arousal [8]. Where a negative perception of error causes burnout in athletes, positive acceptance of mistakes is mentioned as a major factor that increases grit [9], which refers to a person’s perseverance and enthusiasm for overcoming adversity [10]. However, accumulated data on how the role and characteristics of perception of error affect athletes are lacking. Thus, continuously considering the causal model for the perception of error is necessary [11].

The expectancy-value theory can be embedded in this context as a specific variable explained by the perception of error [12,13]. This theory describes the increases in the probability of achievement of individual behavioral characteristics and goals [14], which directly affect one’s motivation for achievement and performance, as well as task selection [14]. This expectancy value is a belief about how well an individual performs a given task. Thus, it is represented by the “expectation” for success, which refers to an individual’s cognitive motivation for the present and future, and “value,” which refers to characteristics that satisfy various needs, such as fun and pleasure, the importance of success, and sacrifice that accompanies training [14]. Furthermore, it is an important factor that promotes the desire and challenge for task performance and changes not only the motivational aspect but also the cognitive and behavioral disposition of an individual [12,15,16].

Immersion and a positive psychological state are formed when the perceived levels of challenge and skill that an individual experiences in a specific activity or task are balanced [17]. This challenge is important because it is closely related to the achievement of goals, expectations of results, and grit. Thus, the challenge acts as a mechanism to promote the attributes related to personal success [17,18].

Siong et al. [19] emphasized expectancy value as an important factor in promoting confidence, challenge, and motivation. Considering this theoretical background, perception of error may have a direct effect on expectancy value and the challenge, and expectancy value may also have a direct effect on the challenge. However, since most existing studies analyzing the relationship between error perception, expectancy value, and challenge reported only a one-dimensional relationship between independent and dependent variables to which regression analysis was applied, cases in which three variables are simultaneously set in the model and analyzed are limited.

Taekwondo athletes are required to select various offensive and defensive playing skills and tactics, and the psychological burden becomes an important factor in the training process for performance improvement and tactical preparation. There is a particularly large room for individual differences in error perception due to the psychological atrophy experienced by young Taekwondo athletes as their performances lead to their future careers. Thus, examining how the perception of error changes the expectancy value and challenge is meaningful, as it can be used as basic data for tactical preparation to improve the psychological and behavioral skills of young Taekwondo athletes.

Accordingly, this study aimed to derive a causal model in which three variables were simultaneously controlled while utilizing the advantages of latent variables by applying structural equation modeling (SEM) [20]. In the investigation of the causal relationship between the three variables, research on the perception of error has focused mainly on the importance of the latter in the field of education, so research in the field of sports is insufficient. Therefore, this study investigated whether error perception could be applied as an antecedent variable for improving athletes’ expectations and challenges. We hypothesized that perception of error would affect the expectancy value and challenge and that the expectancy value would function as a mediating variable in the model.

## 2. Materials and Methods

### 2.1. Participants

This study complied with the STROBE (Strengthening the Reporting of Observational Studies in Epidemiology) checklist. The participants of our study involved middle and high school Taekwondo athletes who participated in the domestic Taekwondo competition in the Republic of Korea. Data were collected from 450 people using convenience sampling. The survey was conducted for nine days, from 10 to 18 May 2020. Of these 450 individuals, data from 423 athletes were used in the final analysis after excluding 27 whose responses were deemed insincere or partially omitted. Prior to the study initiation, informed consent was obtained from all the participants, and the study design was approved by the institutional review board. All procedures and protocols were performed in accordance with ethical standards in studies involving human participants. Additionally, all questionnaire data in this study were processed anonymously for ethical reasons, and the study complied with all coronavirus disease guidelines. The specific demographic characteristics of the participants are listed in Table 1.

### 2.2. Measurement Tools

A structured questionnaire was used as a measurement tool. The detailed questionnaire consisted of 35 questions across four areas as follows: three questions on the demographic characteristics of the participants, seventeen on the perception of error, eight on expectancy value, and seven on challenge. Here, content validity was reviewed by a group of experts (one professor each from a Taekwondo major, a measurement evaluation, and a high school Taekwondo coach). Subsequently, to present evidence of construct validity and internal consistency of each scale, a confirmatory factor analysis (CFA) using maximum likelihood (ML) estimation and reliability analysis using Cronbach’s ɑ coefficient was performed, as summarized in Table 2. The index of fit of the CFA model was determined using the χ^2^ test, Tucker–Lewis Index (TLI, >0.90), comparative fit index (CFI, >0.90), root mean square error of approximation (RMSEA, <0.08), and standardized root mean square residual (SRMR, <0.08) [20], and the responses of all scales were evaluated on a five-point Likert scale.

#### 2.2.1. Perception of Error

In order to measure perception of error, four questions about learning about mistakes, four about challenging mistakes, four about the burden of mistakes, and five about reviewing mistakes were devised based on the Error Orientation Questionnaire developed by Rybowiak et al. [3], which was adapted specifically for young Taekwondo athletes and verified for its quality and validity by Seo et al. [21]. Furthermore, four items (#9, #10, #11, #12) on the burden of mistakes as a subfactor of perception of error were reverse-coded (R). After performing a CFA for perception of error, the fit indices were χ^2^ = 144, df = 62, TLI = 0.968, CFI = 0.975, RMSEA = 0.056, and SRMR = 0.036, and the reliability was >0.80 for all factors (Table 2).

#### 2.2.2. Expectancy Value

In order to measure the expectancy value, four questions, each regarding subjective value and expectations for success, were devised based on the Self- and Task-Perception Questionnaire developed by Eccles et al. [14] that was adapted by Park et al. [22] to suit the Korean culture. The CFA for expectancy value demonstrated that the fit indices were χ^2^ = 70.9, df = 19, TLI = 0.956, CFI = 0.970, RMSEA = 0.080, and SRMR = 0.050, and the reliability was >0.80 for all factors (Table 2).

#### 2.2.3. Challenge

Five single-factor questions to measure the challenge were devised based on the Student Perceptions of Classroom Quality challenge scale developed by Gentry et al. in 2004, which was adapted by Lee et al. [23] to suit the Korean culture and the questionnaire items that Sim et al. [24] verified regarding suitability for athletes. The CFA for the challenge revealed that the fit indices were χ^2^ = 36.8, df = 9, TLI = 0.986, CFI = 0.977, RMSEA = 0.076, and SRMR = 0.018, and the reliability was >0.90 for all factors (Table 2). Thus, the measures of perception of error, expectancy value, and challenge applied in this study were found to satisfy the construct validity, and the reliability in terms of internal consistency was also found to be high.

### 2.3. Data Analysis

The data collected in this study were processed using Jamovi 2.0 (IBM, New York, NY, USA) and AMOS 23.0 (IBM, New York, NY, USA), and the significance level was set to 0.05. Frequency analysis was performed to investigate demographic characteristics, and the CFA and Cronbach’s ɑ coefficient were calculated to examine the validity and reliability of the scales. Furthermore, the basic assumption of normality (skewness ± 3, kurtosis ± 7) [20] was reviewed prior to the SEM, and the hypothesis was verified by sequentially reviewing the measurement and structural models. Correlation analysis was performed to verify the discriminant validity of the perception of errors (learning about mistakes, the burden of mistakes, challenging mistakes, reviewing mistakes), expectation (expectation for success), value (subjective values), and challenge. All correlation coefficients were judged to not overlap the concept of variables when the multicollinearity criterion was less than 0.80 [20]. The bootstrap [25] method was used to verify whether the indirect effect (mediation effect) of the expectancy value in the relationship between perception of error and sense of challenge was statistically significant. At this time, 2000 repetitions were conducted, and statistical significance was confirmed at a bias-corrected 95% confidence interval. Since the mediation effect established in this study corresponds to a multiple mediation model, each indirect effect must be estimated separately. However, only the bootstrapping of the AMOS total indirect effect is presented as an estimate [26]. In order to solve this, the Phantom Model approach was applied as follows: the Phantom Model includes phantom variables in the model to estimate indirect effects and restricts the non-standardized coefficients of the original model to be the same as those of the Phantom Model so that the total effect of this model can be estimated as the indirect effect of the original model (Figure 1). Furthermore, the coefficients of the original model are fixed even if they are converted to the Phantom Model. Hence, even if the Phantom Model is input to the set path model, the fit indices of χ^2^, df, TLI, CFI, RMSEA, and SRMR are equally calculated [26]. For the statistical significance of the mediation effects, the bootstrapping method with a bias-corrected 95% confidence interval was applied [25], and a Phantom Model was established to estimate individual indirect effects [26].

## 3. Results

### 3.1. Normality Test

Since the estimation of the measurement and structural models were set using the ML estimation in this study, normality, which is the basic assumption of ML estimation, was reviewed (Table 3). Hence, skewness was observed to be between −0.085 and 0.304, and kurtosis ranged from −0.964 to 0.618, indicating that the threshold was satisfied [20].

### 3.2. Correlation Analysis

Correlation analysis was performed to verify the discriminant validity between major variables (Table 4). The detailed results of the correlation analysis are as follows: learning about mistakes, a subfactor of perception of error, was positively correlated with expectations for success (r = 0.312, *p* < 0.01), subjective values (r = 0.448, *p* < 0.01), and challenge (r = 0.501, *p* < 0.01). Challenging mistakes, a subfactor in the perception of error, was positively correlated with expectations for success (r = 0.299, *p* < 0.01), subjective values (r = 0.532, *p* < 0.01), and challenge (r = 0.573, *p* < 0.01). Burden of mistakes, a subfactor of perception of error, was positively correlated with subjective values (r = 0.098, *p* < 0.05), although it did not have a significant effect on expectations for success and challenge. Furthermore, reviewing mistakes, a subfactor of perception of error, was positively correlated with expectations for success (r = 0.197, *p* < 0.01), subjective values (r = 0.464, *p* < 0.01), and challenge (r = 0.502, *p* < 0.01). Expectations for success and subjective values, which are subfactors of expectancy value, were positively correlated with challenge (r = 0.701, *p* < 0.01) (r = 0.440, *p* < 0.01), respectively. Moreover, all correlations between major variables were below the multicollinearity criterion of 0.80, indicating no multicollinearity problem [20].

### 3.3. Validation of the Measurement Model

Based on the report by Anderson et al. [27], measurement model validation was performed before structural model validation. The fit indices were as follows: χ^2^ = 238.885, df = 0.98, TLI = 0.959, CFI = 0.966, RMSEA = 0.058, and SRMR = 0.052 (Table 5), indicating a satisfactory fit of the measurement model. The standardization coefficient in which the latent variable explained the measured variable was ≥0.733, suggesting an excellent explanatory power of the measured variable [20].

### 3.4. Validation of the Structural Model

In order to test our hypothesis regarding the causal relationship between perception of error, expectancy value, and challenge, a statistical model with the perception of error as an exogenous variable, expectancy value as an endogenous variable, and challenge as an endogenous and final dependent variable, was constructed (Figure 2), and structural model validation was conducted (Table 6). The fit indices of the model were subsequently as follows: χ^2^ = 238.885, df = 0.98, TLI = 0.959, CFI = 0.966, RMSEA = 0.058, and SRMR = 0.052, indicating that the fit criteria were met [20]. The specific hypothesis test results were as follows: an examination of the effect of perception of error on expectancy revealed that perception of error had a significant positive effect on expectancy (*β* = 0.385, *p* < 0.001). Examination of the effect of perception of error on value revealed that perception of error had a significant positive effect on value (*β* = 0.706, *p* < 0.001). Examination of the effect of perception of error on challenge revealed that perception of error had a significant positive effect on challenge (*β* = 0.367, *p* < 0.001). Examining the result of expectancy and value on challenge revealed that both expectancy and value had significant effects on the challenge (*β* = 0.147, *p* < 0.001) (*β* = 0.460, *p* < 0.001), respectively.

Based on the above results, perception of error was identified as an antecedent variable that promoted expectancy value and challenge, and expectancy value functioned as a mediating variable that could directly or indirectly affect or be affected by the perception of error. Thus, the path effect of the three variables (perception of error, expectancy-error, and challenge) had a significant effect, verifying the statistical significance of the indirect effect (mediation effect).

### 3.5. Validation of Mediation Effect Using Phantom Variables

As a result of analyzing individual indirect effects and statistical significance by applying the Phantom Model in this study (Table 7), expectancy (*p* < 0.003) and value (*p* < 0.001) were both significant in terms of the relationship between perception of error and challenge. Therefore, perception of error may not only directly affect the challenge but also indirectly affect it through a mediating effect on expectations for success and subjective values. Hence, recognizing the perception of error as an accompanying opportunity for growth is important to enhance the expectancy value and challenge in young Taekwondo athletes.

## 4. Discussion

This study set the expectancy value as a mediating variable in the relationship between perception of error and challenge by analyzing data from 423 young Taekwondo athletes to investigate the relationship between the three variables (perception of error, expectancy value, and challenge). We identified perception of error as an antecedent variable that promoted expectancy value and challenge, and the expectancy value was considered a variable that affected and was affected by the perception of error and challenge. Furthermore, perception of error not only had a direct effect on the challenge but also explained the challenge through a mediating effect on the expectation for success and subjective values.

Examining the effect of perception of error on expectancy revealed that the perception of error increased expectations at a significant level. This supports the results of previous studies, which reported that a positive perception of error acts as a mechanism to promote the expectations for success [12,28]. Anshel and Moran [12,29] mentioned the importance of a positive mindset because failure and mistakes are essential experiences to achieving victory. This can be interpreted in the same context as the research results, which argued that mistakes are an accompanying step in athletes’ success. However, young athletes may have difficulty in perceiving mistakes positively because even if the coach emphasizes the importance of a positive perception of error for young athletes, expecting change can be difficult unless the psychological characteristics of the athletes change. Moreover, owing to the competitive nature of sports, being compared to other athletes can lead to a more negative view of mistakes. Thus, coaches should consider both the individual characteristics of the players and the circumstantial aspects for the young athletes to recognize their mistakes as a positive aspect, and efforts should be made to improve the athletes’ perception of error by creating a process-oriented atmosphere rather than a competition-oriented atmosphere.

Examining the effect of perception of error revealed that it increased values at a significant level. This supports the results of previous studies, which reported that perception of error and subjective values were closely related [11,18,30]. Kim et al. [11] reported significant individual differences in value depending on whether the direction of interpreting a mistake was positive or negative, and Weiner [31] suggested that if a mistake was interpreted negatively, the value of continuing with training was also reduced. Therefore, if mistakes are perceived negatively, the value of participation or maintenance of training is reduced, eventually decreasing performance or motivation. Thus, coaches should understand which factors the athletes place more value on and provide informational feedback accordingly. Efforts should further be made to promote the correct values that athletes should pursue by providing feedback that incorporates the values pursued, focusing on individual skill proficiency rather than solely on victory. A learning approach toward acceptance of mistakes rather than avoidance should be encouraged

Examining the effect of perception of error on the challenge revealed that perception of error increased the challenge at a significant level. This supports the results of previous studies, which reported that a positive perception of error acted as a mechanism to promote the desire for challenge [17,24,32]. Dweck et al. [33] mentioned the importance of perceiving mistakes positively to promote the desire for challenge. Since the latter acts as a factor in alleviating athletes’ slumps, it can be interpreted in the same context as the research results of Gray [34], who argued the importance of perceiving mistakes as an opportunity for growth. Hence, it is important for athletes to perceive their mistakes positively to cultivate a desire for challenge [35]. This is because the perception of error is known to be deeply related to self-efficacy, which is closely related to the challenge. Increasing the athletes’ self-efficacy can be interpreted as an important factor that can improve the perception of error in a positive direction. Therefore, competent coaching ability is required to reduce the burden of mistakes through the experience of success by presenting an appropriate level of task difficulty to the athletes and cultivating a desire for challenge by encouraging an active attitude.

Examining the effect of expectation on the challenge revealed that expectation increased the challenge at a significant level. This supports the results of previous studies, which reported that positive expectations for success served as a psychological mechanism to foster a challenging attitude toward tasks [13,18,36]. Butler et al. [37] reported that expectations for success and challenging attitude were very closely related to each other and could be interpreted in the same context as the research results of Bali et al. [38], who argued that constant challenge was important to achieve one’s selected goals in competitive sports. Furthermore, aiming for a positive attitude is important because the expectation for success can give meaning to an individual’s experiential domain and promote motivation for participation in tasks [39,40]. The expectation for success can therefore induce the intrinsic motivation of Taekwondo athletes, thereby encouraging a challenging attitude toward a task. Therefore, coaches must identify factors that lower the psychological attributes through systematic consultation with the athletes in advance, to allow them to form positive psychological attributes and training methods for increasing their performance. Young athletes need the ability to examine their own mistakes through video analysis of training and game situations.

Examining the effect of value on the challenge demonstrated that value increased the challenge at a significant level. This supports the results of previous studies, which reported that value acted as a factor in increasing intrinsic motivation [41,42]. This result can be interpreted in the same context as the research results of Cox et al. [15], who reported that athletes’ sense of identity acted as a factor promoting various positive values (performance, interest, and usefulness). Thus, the athletes’ sense of identity fosters positive values, through which they can persevere even in challenging situations.

Considering the path effect of expectations and values, they could be considered the main factors in determining athletes’ challenge. Weiss [43] argued that these two factors determined the level of commitment of athletes. Thus, it is important for coaches to help athletes establish positive expectations and correct values for performance. This study, in which expectancy and value were identified as significant variables in the relationship between perception of error and challenge, also supports this result. Moreover, perception of error had a particularly greater indirect effect on the challenge when mediated by values compared to when mediated by expectations. This suggests that the process by which perception of error affects the challenge, the value of the task to be challenged, and whether it arouses interest to play more important roles than the expectation of being able to complete the task well.

Our study had the following limitations: although the sample of this study was representative of youth Taekwondo players, convenience sampling had to be carried out due to difficulties in running and conducting the competition. In a follow-up study, it is necessary to use random sampling to reflect the general characteristics of youth Taekwondo players’ careers. In addition, although the perception of errors explained expectation, value, and challenge, this study did not consider variables that could reinforce the perception of errors or conduct interviews with coaches or players. Due to the limited discussion on how to increase the recognition of mistakes, it is necessary to consider various variables that explain it in follow-up studies while conducting qualitative research that can determine them phenomenologically.

## 5. Conclusions

The following hypotheses were verified by applying SEM based on the collected data: first, examining the effect of perception of error on expectancy confirmed that the perception of error had a positive effect on expectancy. Second, perception of error had a positive effect on value. Third, examining the effect of perception of error on the challenge confirmed that perception of error had a positive effect on the challenge. Fourth, examining the effect of expectancy on the challenge confirmed that expectancy had a positive effect on the challenge. Fifth, examining the effect of values on the challenge confirmed that values had a positive effect on the challenge. Last, the expectancy value had a significant mediation effect on the relationship between the perception of error and challenge. Therefore, a positive perception of error not only raises expectancy value but also enhances the challenge.

## Figures and Tables

**Figure 1 ijerph-19-06112-f001:**
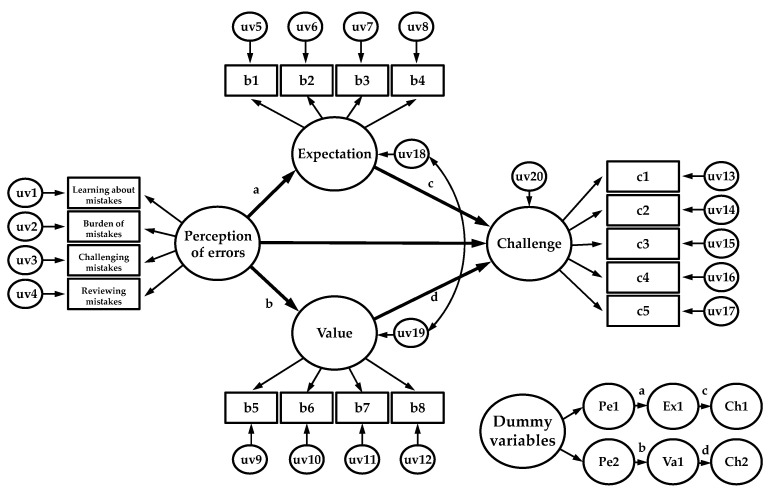
Structural equation modeling (SEM) analysis results. uv = unobserved variables, Pe = perception of errors, Ex = expectation, Va = values.

**Figure 2 ijerph-19-06112-f002:**
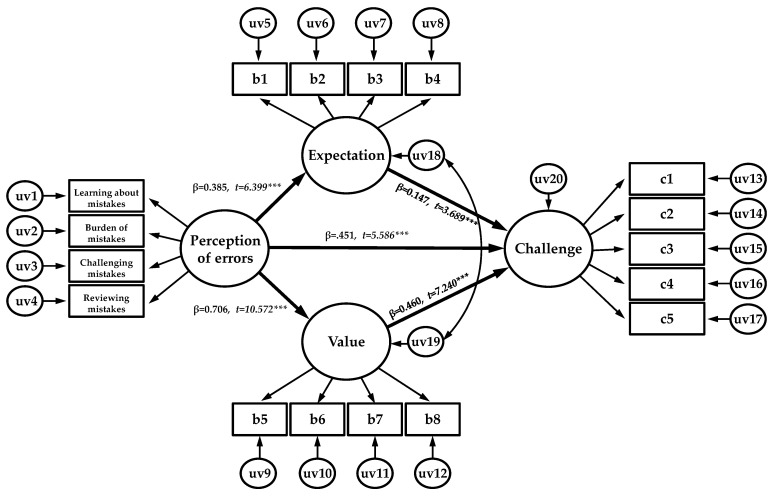
Structural equation modeling (SEM) analysis results. χ^2^ = 238.885, df = 98, TLI = 0.959, CFI = 0.966, RMSEA = 0.058, SRMR = 0.052. *** *p* < 0.001. uv = unobserved variables.

**Table 1 ijerph-19-06112-t001:** Characteristics of the participants.

Demographics	Category	Frequency	N (%)
Sex	Male	290	68.6
	Female	133	31.4
Affiliation	Middle school	212	50.1
	High school	211	49.9
Career	Less than three years	190	44.9
	More than three years to less than six years	110	26.0
	More than six years	123	29.1
National competition award	Yes	257	60.8
	No	166	39.2
Total		423	100

**Table 2 ijerph-19-06112-t002:** Confirmatory factor analysis and reliability analysis of perception of errors, expectancy value, and challenge.

Latent Variable		Variable		B	*β*	S.E	t	α	χ2	df	TLI	CFI	RMSEA	SRMR
Perception of errors	Learning about mistakes	→	a1	0.805	0.840	0.038	20.9 ***	0.919	179	113	0.980	0.984	0.037	0.033
		→	a2	0.820	0.882	0.036	22.6 ***							
		→	a3	0.825	0.884	0.036	22.7 ***							
		→	a4	0.793	0.813	0.039	19.9 ***							
	Burden of mistakes	→	a5	0.811	0.771	0.045	17.8 ***	0.842						
		→	a6	0.911	0.852	0.044	20.5 ***							
		→	a7	0.799	0.805	0.042	18.9 ***							
		→	a8	0.646	0.698	0.041	15.5 ***							
	Challenging mistakes	→	a9	0.616	0.687	0.040	15.0 ***	0.843						
		→	a10	0.777	0.754	0.045	17.1 ***							
		→	a11	0.749	0.784	0.041	18.0 ***							
		→	a12	0.777	0.738	0.047	16.5 ***							
	Reviewing mistakes	→	a13	0.600	0.714	0.037	16.2 ***	0.903						
		→	a14	0.669	0.785	0.036	18.5 ***							
		→	a15	0.694	0.791	0.037	18.8 ***							
		→	a16	0.717	0.820	0.036	19.8 ***							
		→	a17	0.719	0.790	0.038	18.7 ***							
Expectancy	Expectation for success	→	b1	0.693	0.725	0.042	16.4 ***	0.854	70.9	19	0.956	0.970	0.080	0.050
		→	b2	0.643	0.663	0.044	14.5 ***							
		→	b3	0.786	0.864	0.037	21.0 ***							
		→	b4	0.806	0.843	0.039	20.3 ***							
Value	Subjective values	→	b5	0.732	0.751	0.042	17.3 ***	0.871						
		→	b6	0.800	0.842	0.039	20.4 ***							
		→	b7	0.715	0.840	0.035	20.3 ***							
		→	b8	0.685	0.795	0.366	18.7 ***							
Challenge		→	c1	0.721	0.800	0.033	21.6 ***	0.914	36.8	9	0.986	0.977	0.076	0.018
		→	c2	0.763	0.877	0.030	24.9 ***							
		→	c3	0.707	0.820	0.031	22.4 ***							
		→	c4	0.775	0.855	0.032	23.9 ***							
		→	c5	0.688	0.754	0.034	19.8 ***							

*** *p* < 0.001.

**Table 3 ijerph-19-06112-t003:** Normality test.

Contents	Skewness		Kurtosis	
	S	Structural Equation Modeling (SEM)	S	SEM
Learning about mistakes	−0.450	0.119	0.315	0.237
Challenging mistakes	−0.147		−0.366	
Burden of mistakes	−0.389		0.095	
Reviewing mistakes	0.112		−0.344	
Expectations for success	−0.067		0.314	
Subjective values	−0.604		0.197	
Challenge	−0.231		−0.048	

**Table 4 ijerph-19-06112-t004:** Analysis of correlation between variables.

Contents	1	2	3	4	5	6	7
Learning about mistakes	1						
Challenging mistakes	r = 0.609, *p* < 0.01	1					
Burden of mistakes	r = 0.128, *p* < 0.01	r = 0.041	1				
Reviewing mistakes	r = 0.407, *p* < 0.01	r = 0.474, *p* < 0.01	r = 0.033	1			
Expectations for success	r = 0.312, *p* < 0.01	r = 0.299, *p* < 0.01	r = 0.066	r = 0.197, *p* < 0.01	1		
Subjective values	r = 0.448, *p* < 0.01	r = 0.532, *p* < 0.01	r = 0.098, *p* < 0.05	r = 0.464, *p* < 0.01	r = 0.366, *p* < 0.01	1	
Challenge	r = 0.501, *p* < 0.01	r = 0.573, *p* < 0.01	r = 0.075	r = 0.502, *p* < 0.01	r = 0.701, *p* < 0.01	r = 0.440, *p* < 0.01	1

**Table 5 ijerph-19-06112-t005:** Measurement model path.

Latent Variable	m Variable	B	*β*	t
Perception of error	Learning about mistakes	1.000	0.717	
	Challenging mistakes	1.061	0.806	13.740 ***
	Burden of mistakes	0.825	0.674	10.948 ***
	Reviewing mistakes	0.737	0.619	11.187 ***
Expectations	b1	1.000	0.859	
	b2	1.033	0.844	19.563 ***
	b3	0.887	0.725	16.316 ***
	b4	0.828	0.668	14.657 ***
Values	b5	1.000	0.755	
	b6	1.058	0.820	17.024 ***
	b7	0.970	0.841	17.492 ***
	b8	0.947	0.810	16.819 ***
Challenge	c1	1.000	0.847	
	c2	0.979	0.842	21.754 ***
	c3	0.959	0.886	23.717 ***
	c4	0.901	0.809	20.381 ***
	c5	0.933	0.823	20.929 ***

χ^2^ = 238.885, df = 0.98, TLI = 0.959, CFI = 0.966, RMSEA = 0.058, SRMR = 0.052. *** *p* < 0.001.

**Table 6 ijerph-19-06112-t006:** Structural equation modeling (SEM) analysis result.

Latent Variable				B	*β*	S.E	t	Hypothesis View
Hypothesis 1	Perception of error	→	Expectations of success	0.495	0.385	0.077	6.399 ***	Selected
Hypothesis 2	Perception of error	→	Subjective values	0.855	0.706	0.081	10.572 ***	Selected
Hypothesis 3	Perception of error	→	Challenge	0.451	0.367	0.078	5.586 ***	Selected
Hypothesis 4	Expectations of success	→	Challenge	0.141	0.147	0.037	3.689 ***	Selected
Hypothesis 5	Subjective values	→	Challenge	0.466	0.460	0.063	7.240 ***	Selected

χ^2^ = 238.885, df = 98, TLI = 0.959, CFI = 0.966, RMSEA = 0.058, SRMR = 0.052. *** *p* < 0.001.

**Table 7 ijerph-19-06112-t007:** Individual indirect effect verification results using the Phantom Model.

Mediating Path					Indirect Effect	*p*
Perception of error	→	Expectations	→	Challenge	0.070	0.003
Perception of error	→	Values	→	Challenge	0.399	0.001

## Data Availability

The authors declare that all data and materials are available to be shared on a formal request.

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
