# Peer review of "Importance of Perception of Errors and Challenges for Improving Psychological Conditioning: Mediating Effect of Expectancy-Value Using the Phantom Model for Taekwondo Athletes"

_ijerph, 2022, doi:10.3390/ijerph19106112_

Round 1
Reviewer 1 Report
Dear Editors,
Have a nice day.
This study analyzes the causal relationship between the perception of error, expectancy value, and task challenge in 423 young Taekwondo athletes. The study is technically fine and meets the basic merits to warrant publication. However, the title of this study is a bit vague. This study is related to athletes and sports. I suggest authors add 'athletes' or 'sports' keywords in the title and the keywords section to specify the nature of this study and mention related conclusions with keywords in the conclusion section as well. Furthermore, the reading of this paper is not very smooth to conceptualize. Moderate English editing is required. Rest is fine from my side.
Author Response
Response to Reviewers
Manuscript Title: Importance of perception of errors and challenges for improving psychological conditioning: Effect of multiple mediation using the Phantom Model
Editorial Board Member,
International Journal of Environmental Research and Public Health
We appreciate the constructive comments of the reviewers. We are returning herewith the above manuscript, which has been thoroughly revised as per your letter. We hope that the revised manuscript will be accepted for publication in International Journal of Environmental Research and Public Health.
To Reviewers:
We would like to thank the two reviewers for their critical comments and suggestions for our manuscript. We have read each reviewer’s comments with great care and revised the manuscript accordingly. Below we address each comment point by point. Thus, we believe that the revised version of our manuscript has been substantially improved compared to the initial submission.

Reviewer 2 Report
Authors of this article set out to analyse the mediation of the expectation value in the relationship between the perception of error and the challenge posed by an activity, developing a model called the Phantom Model. This model is useful in those cases in which you try to relate so many variables at the same time. The authors focus on young middle and high school taekwondo athletes.
It is recommended to review some points before publication in this journal:
- Page 3, line 103: This section refers to the demographic variables collected from the sample of participants. Age is not specifically referenced, but whether they are middle school or high school is recorded. It would be interesting to specifically include this variable. On the other hand, the length of his career in this sport is recorded. Has the statistical analysis been carried out taking these variables into account? In a certain way, these two variables determine how the individual faces the challenge, and his perspective of the mistakes made, considering the previous experiences of the subject. It would be convenient to specifically review the role of these variables in these relationships in order to specifically conclude the variables on which to influence, considering the age and the time they have been practicing this sport. If this analysis is not going to be performed, it should be included in the discussion section as a limitation of the study.
- Page 5, line 146: This section does not specify the statistical strategy carried out to perform the correlation of the variables in section 3.2 (page 7, line 181).
- Page 7, line 188: When establishing the correlation between burden of mistakes, a positive correlation is established with the subjective values (r=.098, p<.05). Despite this significant analysis, the value of r must be considered, and it is close to zero, so it is likely that there is no linear relationship between the variables.
- Page 7, table 4: This table represents the correlations between some of the study variables. On the one hand, the cells represent the value of r, accompanied by asterisks that reflect the statistical significance. This could give rise to interpretation error, since, at a quick glance, the reader would observe a value of 0.573 as significant between the challenging errors and the challenge for example, being able to think that this p value is not significant. It is recommended that the value of r be included in each cell, specifically accompanied by its p value (r=.573, p<.01).
- Tables in general: The numbering and position of the tables should be reviewed. In the text, after referencing table 1 (page 3, line 104), table 3 (page 3, line 114) is referenced. It is sometimes necessary to specify the meaning of the acronyms they contain in the table footer. On the other hand, in tables 1 and 2, lines are needed to separate the rows that indicate the sections of the variables studied in them.
- Page 3, line 93: It is necessary to specify how the data has been collected, the means through which the questionnaires were carried out and the procedures carried out to avoid duplicating records, as well as the selection techniques of the study sample since it is only reported that the sampling was for convenience.
- Page 10, line 240: It would be convenient to include a paragraph on limitations of the study that could be causing some bias in the observed results.
Best regards
Author Response

(The authors gave the same response as above.)

Reviewer 3 Report
The summary of the article is well written, and it is clear how the research was developed, as well as the size of the sample, and some of the conclusions that this research allowed to be drawn. However, the Abstract could be improved if the authors would indicate the Research Methodology they followed.
The Literature Review of the paper is extensive and has been well developed, and addresses the themes proposed for the research. However, as a recommendation, the authors should have been more careful in the selection of the selected articles, as there are some articles which are already some years old (1988, 2002, 1999, 2004, etc.). I believe that they are probably some authors with some reference, however I also believe that some authors have recently applied these concepts to more current situations...
[line 97] The authors indicate that they selected 450 athletes, however the authors do not indicate why this sample size was chosen, nor did they identify what the criteria were for the selection of this sample. To be part of this sample, is it enough to be a Taekwondo athlete?
It would be very important for the validity of the research, that the authors indicate how the questionnaire was validated. How was the validation done? What validation criteria were taken into consideration?
[line 253] Why do the authors put "... Farson et al. [29] ...", being only doia authors in this reference? Why did the authors not put "... Farson and Keyes [29] ..."? The authors should review and correct this situation.
Did the research developed by the authors have limitations? If the authors feel that this study had some limitations, which led to this research not having the desired level, the authors should put this information in the article, and share this information with the scientific community.
It would be very beneficial to the scientific community if, at the end of the article, the authors left recommendations for future works. In other words, they should recommend research works that could be developed by other researchers in the future, so that the scientific community could give continuity to the research developed by the authors in this article.
Author Response

(The authors gave the same response as above.)

Round 2
Reviewer 3 Report
The improvement work carried out by the authors allowed increasing the scientific level of the article.